# Cross Talk between GlAQP and NOX Modulates the Effects of ROS Balance on Ganoderic Acid Biosynthesis of *Ganoderma lucidum* under Water Stress

Quanyu Zhu,[a] Ang Ren,[a] Juan Ding,[a] Jian He,[a] Mingwen Zhao,[a] Ailiang Jiang,[a] Xiaolin Zhou,[a] Jieying Wang,[a] Qin He[a]

[a]Sanya Institute of Nanjing Agricultural University, Department of Microbiology, Key Laboratory of Agricultural Environmental Microbiology, Ministry of Agriculture, College of Life Sciences, Nanjing Agricultural University, Nanjing, Jiangsu, People's Republic of China

**ABSTRACT** Water stress affects both the growth and development of filamentous fungi; however, the mechanisms underlying their response to water stress remain unclear. In this study, water stress was found to increase intracellular reactive oxygen species (ROS) level, ganoderic acid (GA) content, and NADPH oxidase (NOX) activity of *Ganoderma lucidum* by 148.45%, 75.32%, and 161.61%, respectively. Water stress induced the expression of the *G. lucidum* aquaporin (GlAQP) gene, which facilitated water transfer for microbial growth. Compared to wild type (WT), exposure to water stress increased growth inhibition rate, ROS level, and GA content of *GlAQP*-silenced strains by 37 to 41%, 36 to 38%, and 25%, respectively. Furthermore, at the early stage of fermentation in *GlAQP*-silenced strains, water stress resulted in 16 to 17% and 9 to 10% lower ROS level and GA content compared to WT, respectively. However, in *GlAQP*-overexpressing strains, ROS level and GA content were 22 to 24% and 12 to 13% higher than in WT, respectively. In *GlAQP*-silenced strains, water stress at the late stage resulted in 35 to 37% and 29 to 30% higher ROS level and GA content, respectively, while in *GlAQP*-overexpressing strains, levels were 16 to 17% and 9% lower than WT, respectively. Cross talk between GlAQP and NOX positively regulated the GA biosynthesis of *G. lucidum* via ROS under water stress at the early stage but this regulation became negative at the late stage. This study deepens the understanding of fungal signaling transduction under water stress and provides a reference for analyzing environmental factors that influence the regulation of the fungal secondary metabolism.

**IMPORTANCE** *Ganoderma lucidum* is an advanced basidiomycete that produces medicinally active secondary metabolites (especially ganoderic acid [GA]) with high commercial value. Water stress imposes an important environmental challenge to *G. lucidum*. The mechanism of GA biosynthesis under water stress and the role of *G. lucidum* aquaporin (GlAQP) during its biosynthesis remain unclear. Moreover, the effect of the relationship between GlAQP and NADPH oxidase (NOX) on the level of reactive oxygen species and GA production under water stress is unknown. This study provides information on the biological response mechanism of *G. lucidum* to water stress. A new theory on the cell signaling cascade of *G. lucidum* tolerance to water stress is provided that also incorporates the biosynthesis of secondary metabolites involved in NOX and GlAQP.

**KEYWORDS** water stress, cross talk, ROS, aquaporin, NADPH oxidase, ganoderic acid

Address correspondence to Qin He, qhe@njau.edu.cn.

The authors declare no conflict of interest.

Water is indispensable for organisms and a lack of water affects various physiological activities in cells and whole organisms. Water stress, which refers to a water deficit, is one of the most common abiotic stresses and is currently one of the most serious environmental pressures affecting agriculture. Microorganisms are more susceptible to water stress because of their limited ability to escape adverse environmental

conditions. Thus, physiological changes occur commonly. Dehydration has been shown to cause oxidative stress and lipid peroxidation damage in yeast cells (1). The structure and function of many organelles change, and the intracellular protective response is activated (2). Water stress affects the growth rate, biomass, and plant hormone content of ectomycorrhizal fungi (3). In desert truffles, moderate water stress significantly increases alkaline phosphatase content, which is an adaptation to drought (4). In addition, water stress has an important influence on germination, germ extension, growth rate, internal cell water potential, and the accumulation of sugar and sugar alcohol in *Fusarium graminearum* (5).

Water stress usually induces an excessive production of reactive oxygen species (ROS), which change the redox balance of cells and causes oxidative stress and oxidative damage in cells (6). Organisms have evolved antioxidant defense systems to reduce the oxidative damage caused by ROS (7). However, it has been shown that ROS play an important signaling role under abiotic stress, as they trigger defense and adaptive responses (8). The mechanism regulating ROS signaling under water stress has been deeply studied in plants. Shi et al. (9) reported that *OsRbohB*-mediated ROS generation plays an important role in the water stress tolerance of rice. However, the response mechanism of filamentous fungi to water stress has not been explored in depth, and the exact role of ROS in signal transduction under water stress remains unclear.

Aquaporins (AQPs) are a class of transmembrane channel proteins belonging to the conserved superfamily of major intrinsic proteins (MIPs). In addition to transporting water, AQPs can also transport glycerol, urea, $CO_2$, $NH_3$, and $H_2O_2$ (10). Many studies have shown that AQPs respond to drought and help plants cope with water stress (11). Cui et al. (12) found that expression of the broad bean AQP gene *VfPIP1* in *Arabidopsis* can improve the drought resistance of transgenic plants by promoting stomata closure in response to water stress. However, stomata closure also reduces plant photosynthesis. Because AQPs have the function of transporting water and $CO_2$, they play an important role in the process of plant photosynthesis (13).

In yeast, AQPs are important to establish freezing resistance, spore formation, and morphological adaptation (14). Compared with studies on plants and yeasts, studies on the role of AQPs in filamentous fungi are rare. AQPs play an important role in the pH-dependent germination of *Rhizopus delemar* spores (15). Regarding the function of AQPs in filamentous fungi under water stress, this is the only study showing that the AQPs of the arbuscular mycorrhizal fungus *Glomus intraradices* can transport water from this fungus to host plants. This transport improves the drought resistance of host plants (16). Further in-depth research on the function of AQP in filamentous fungi under water stress is warranted.

*Ganoderma lucidum* is a medicinal fungus that has been used for thousands of years in Asia. It has a variety of pharmacologically active ingredients that can be used to treat cancer, regulate immunity, and promote health and longevity (17). Of these active ingredients, ganoderic acid (GA) is one of the main secondary metabolites with pharmacological activity. Previous research mainly focused on the separation and purification of active ingredients and the optimization of fermentation conditions, while research on the biosynthesis of GA is rare. Aided by the sequencing of the complete of *G. lucidum* genome, genetic transformation, and the development and application of gene-silencing systems, further basic biological research has become possible (18–20). Recent studies found that exogenous addition of methyl jasmonate induces an increase in GA biosynthesis via ROS (21). In addition, environmental factors also affect GA biosynthesis. GA biosynthesis increases under heat stress, and $Ca^{2+}$, ROS, and NO signaling all play regulatory roles in GA biosynthesis (22–24).

In this research, the mechanism of GA biosynthesis under water stress was studied with a specific focus on the role of *G. lucidum* aquaporin (GlAQP) in GA biosynthesis. Moreover, the effect of the relationship between GlAQP and NADPH oxidase (NOX) on ROS levels and GA production was assessed at both early and late stages of water

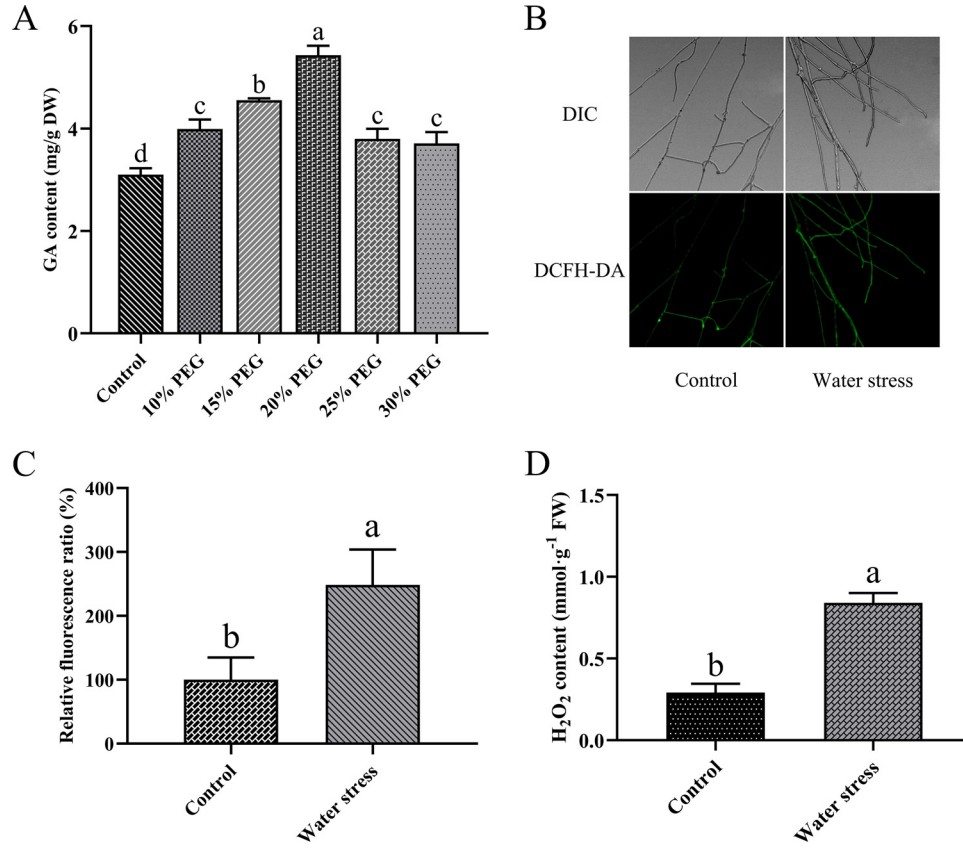

**FIG 1** Effect of water stress on ROS level and GA content of *G. lucidum*. (A) The GA content in WT treated with different concentrations of PEG. (B) Change in ROS level detected by DCFH-DA staining in WT under water stress. (C) Change in ROS fluorescence ratio in WT under water stress. (D) $H_2O_2$ content in WT under water stress. DW, dry weight; FW, fresh weight. The values indicate the mean ± SD of three independent experiments. Different letters indicate significant differences between treatments (Duncan's multiple range test, $P < 0.05$; Student's *t* test, $P < 0.05$).

stress. This study contributes to a better understanding of the biological response mechanism of *G. lucidum* to water stress and provides new insights on the cellular signaling cascades involved in the biosynthesis of *G. lucidum* secondary metabolites.

## RESULTS

**Water stress inhibits growth of *G. lucidum* mycelia and increases GA content.** To determine the effect water stress has on the growth of *G. lucidum*, polyethylene glycol (PEG) was added to the culture medium (25). The growth of *G. lucidum* on solid CYM medium containing different PEG concentrations was assessed. As shown in Fig. S1A and S1B in the supplemental material, the growth of *G. lucidum* on CYM solid medium was increasingly inhibited with increasing PEG concentration. GA has important economic and pharmacological value, and its content is one of the criteria used to judge the quality of *G. lucidum*. The results showed that with increasing PEG concentration, the GA content increased first and then decreased, reaching its maximum at 20% PEG addition to the medium. Compared with the control, GA contents increased by 28.86%, 46.91%, 75.32%, 22.61%, and 19.73% in response to PEG concentrations of 10%, 15%, 20%, 25%, and 30%, respectively (Fig. 1A). Under 20% PEG treatment, compared to the control, the transcription level of the three key enzyme genes of the GA biosynthesis pathway *hmgr* (encoding 3-hydroxy-3-methylglutaryl-CoA reductase), *sqs* (encoding squalene synthase), and *osc* (encoding lanosterol synthase) increased by 158.14%, 161.63%, and 187.33%, respectively (Fig. S1C). Therefore, 20% PEG was added to the medium to induce water stress in the following experiments.

**ROS play an important role in GA biosynthesis under water stress.** Previous studies found that ROS can regulate GA biosynthesis (21, 23), and water stress often causes an increase of ROS (6). In this paper, the hypothesis is put forward that the increase of GA biosynthesis under water stress may be regulated by ROS. To test this hypothesis, first, 2′,7′-dichlorodihydrofluorescein diacetate (DCFH-DA) was used as a ROS fluorescent probe to detect the intracellular ROS level of *G. lucidum*. Fluorescence analysis showed that compared with the control, under water stress, the fluorescence intensity of DCFH-DA in *G. lucidum* cells increased by 148.45% (Fig. 1B and C). Next, the $H_2O_2$ content was measured. Compared with the control, under water stress, the intracellular $H_2O_2$ content of *G. lucidum* increased by 187.51% (Fig. 1D). In addition, the activity of NOX (a key enzyme that produces ROS in *G. lucidum*) and the activities of antioxidant enzymes, such as catalase (CAT), superoxide dismutase (SOD), and glutathione peroxidase (GPx), were assessed. The results show that compared with the control, under water stress, NOX activity increased by 161.61%, while the activity of antioxidant enzymes such as CAT, SOD, and GPx increased by 63.39%, 57.30%, and 94.41%, respectively (Fig. S1D). These results indicate that water stress increases the ROS level and the activity of related enzymes in the ROS production and scavenging system.

To explore the role ROS play in GA biosynthesis under water stress, the effect of the two ROS scavengers *N*-acetyl-L-cysteine (NAC) and vitamin C (VC), also known as L-ascorbic acid, on GA production was tested. As a positive control, 8 mM $H_2O_2$ was added. Compared with the control, the addition of exogenous 8 mM $H_2O_2$ under water stress increased the ROS level by 137.75%, and the intracellular ROS level by 148.92%. Exogenous addition of 0.5 mM NAC and 1 mM VC under water stress decreased ROS levels by 57.53% and 57.21%, respectively (Fig. 2A and B). Compared with untreated strains, under water stress, the $H_2O_2$ content increased by 181.11% in response to the exogenous addition of 8 mM $H_2O_2$, and the intracellular $H_2O_2$ content increased by 183.97%. Exogenous addition of 0.5 mM NAC and 1 mM VC under water stress reduced the intracellular $H_2O_2$ contents by 43.57% and 41.77%, respectively (Fig. 2C). Compared with the control, the addition of 8 mM $H_2O_2$ under water stress increased GA biosynthesis levels by 75.85% and 76.87%, respectively. Exogenous addition of 0.5 mM NAC and 1 mM VC reduced water-stress-induced GA biosynthesis by 43.27% and 43.41%, respectively (Fig. 2D). After addition of 0.5 mM NAC and 1 mM VC, the transcription level of key enzyme genes in the GA biosynthetic pathway decreased by 58.57% and 65.68% (*hmgr*), 69.13% and 58.99% (*sqs*), as well as 72.36% and 65.56% (*osc*), respectively, compared with water stress (Fig. S1E). These results indicate that the accumulation of ROS induced by water stress promoted GA biosynthesis.

**Effect of water stress on the transcription level of AQP genes in *G. lucidum* and cloning of the *GlAQP* gene.** It has been reported that AQP plays an important role in the coping mechanism of plants exposed to water stress (12, 13). To study the functions of GlAQPs under water stress, a total of five AQP genes were identified, and their expression levels under water stress were studied. As shown in Fig. S2A, the GlAQP gene *GL23609-R1* showed the most significant response to water stress. Its expression level was 3.05 times higher than that of the control, while the transcription levels of other AQP genes were downregulated. Therefore, this paper focuses on studying the function of the GlAQP gene *GL23609-R1* under water stress, which was named *GlAQP*. The *GlAQP* gene has a length of 951 bp, and the encoded protein contains 316 amino acids. Domain analysis showed that the GlAQP protein contains major intrinsic protein conserved domains and six transmembrane domains. Subcellular localization prediction indicated that GlAQP was localized on the cell membrane. Amino acid sequence alignment showed that the GlAQP protein contains two asparagine-proline-alanine motifs (Fig. S2B), and phylogenetic analysis showed that the GlAQP protein belongs to the aquaglyceroporin channel protein family (Fig. S2C).

**_GlAQP_ silencing increases the sensitivity of _G. lucidum_ to water and oxidative stress.** To explore the role of GlAQP in *G. lucidum* coping with water stress, the RNAi-silencing vector pAN7-dual-GlAQPi was constructed and transformed into *G. lucidum* to construct *GlAQP*-silenced strains. Twenty-eight positive transformants were screened, the *GlAQP* gene expression levels of which were detected by quantitative real-time PCR. A

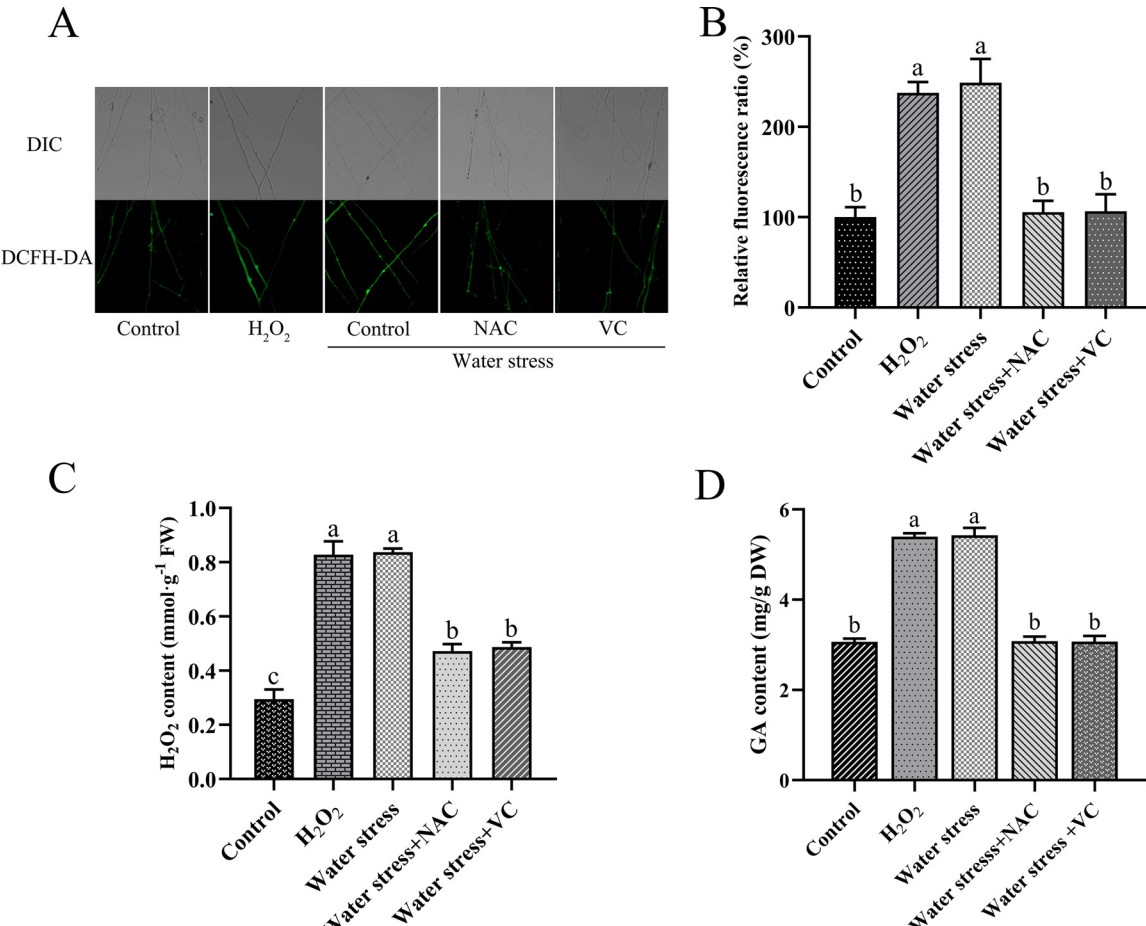

**FIG 2** Effect of ROS scavengers on ROS level and GA content of *G. lucidum* under water stress. (A) Change in ROS level detected by DCFH-DA staining in WT treated with 8 mM $H_2O_2$, 0.5 mM NAC, or 1 mM VC under water stress. (B) Change in ROS fluorescence ratio in WT treated with 8 mM $H_2O_2$, 0.5 mM NAC, or 1 mM VC under water stress. (C) $H_2O_2$ content in WT treated with 8 mM $H_2O_2$, 0.5 mM NAC, or 1 mM VC under water stress. (D) The GA content in WT treated with 8 mM $H_2O_2$, 0.5 mM NAC, or 1 mM VC under water stress. The values indicate the mean $\pm$ SD of three independent experiments. Different letters indicate significant differences between treatments (Duncan's multiple range test, $P < 0.05$).

total of 12 transformants with a silencing efficiency of 80 to 90% were screened. The *GlAQP*-silenced strains GlAQPi4 and GlAQPi9 were randomly selected for further study (Fig. S3A). Compared with the wild type (WT), under water stress, the growth inhibition rates of GlAQPi4 and GlAQPi9 increased by 40.20% and 37.23%, respectively. These results show that *GlAQP*-silenced strains were more sensitive to water stress (Fig. S3B and S3C). Water stress often leads to an overproduction of ROS, which causes oxidative stress (6). Previous studies have shown that plant AQPs can reduce ROS accumulation under water stress and enhance the drought resistance of plants (26). The results of the present study are consistent with previous research. In this study, *GlAQP* responded to oxidative stress. When exposed to 8 mM $H_2O_2$, the transcription level of *GlAQP* increased significantly and was 3.91 times higher than that of the control (Fig. S3D). The sensitivity of *GlAQP*-silenced strains to $H_2O_2$ also significantly increased. The growth inhibition rates of GlAQPi4 and GlAQPi9 increased by 71.45% and 68.54%, respectively, compared with WT in response to treatment with 8 mM $H_2O_2$ (Fig. S3B and C). This result indicates that silencing *GlAQP* reduces the tolerance of *G. lucidum* to oxidative stress.

**GlAQP affects GA biosynthesis via ROS under water stress.** The GA content was assessed to evaluate whether GlAQP affects GA biosynthesis. Under water stress, the GA contents in GlAQPi4 and GlAQPi9 were 25.22% and 25.15% higher than in WT, respectively (Fig. 3A). Compared with WT, in GlAQPi4 and GlAQPi9, the transcription

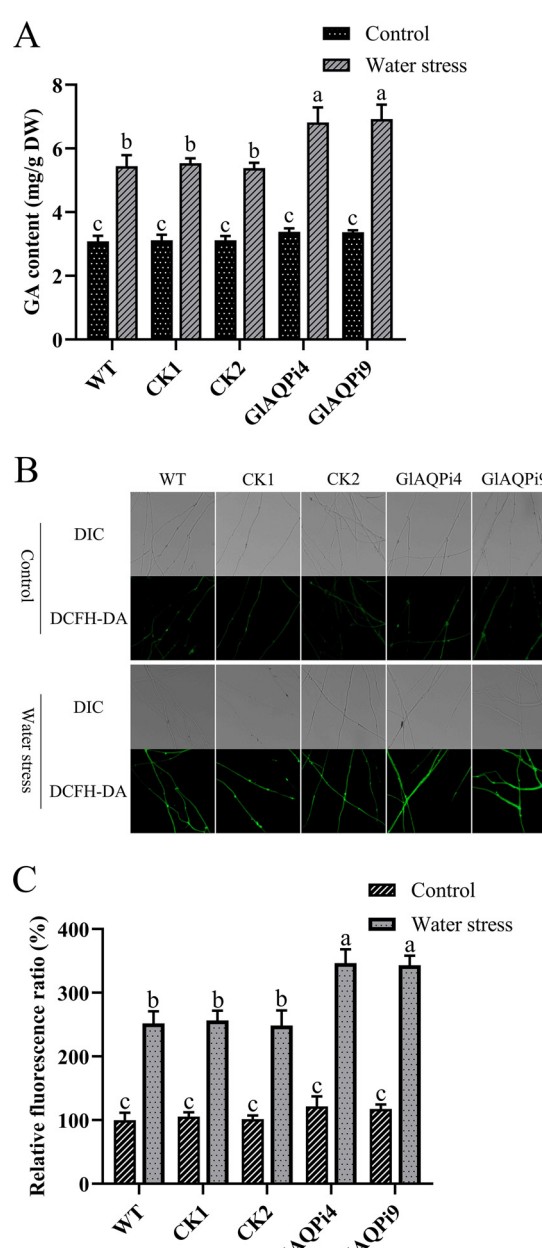

**FIG 3** Effect of silencing *GlAQP* on ROS level and GA content of *G. lucidum* under water stress. (A) The GA content in *GlAQP*-silenced strains under water stress. (B) Change in ROS level detected by DCFH-DA staining in *GlAQP*-silenced strains under water stress. (C) Change in ROS fluorescence ratio in *GlAQP*-silenced strains under water stress. The values indicate the mean ± SD of three independent experiments. Different letters indicate significant differences between treatments (Duncan's multiple range test, $P < 0.05$).

level of key enzyme genes in the GA biosynthesis pathway also increased by 31.06% and 32.65% (*hmgr*), 30.99% and 31.92% (*sqs*), as well as 39.31% and 40.76% (*osc*), respectively (Fig. S4A to C).

Because of the increased sensitivity of *GlAQP*-silenced strains to oxidative stress, it was further explored whether GlAQP can regulate the ROS level of *G. lucidum* under water stress. The intracellular ROS levels of *GlAQP*-silenced strains were assessed first. In GlAQPi4 and GlAQPi9, the levels were 37.52% and 36.19% higher than in WT, respectively (Fig. 3B and C). The intracellular $H_2O_2$ contents of *GlAQP*-silenced strains were also tracked. Compared with WT, the intracellular $H_2O_2$ contents of GlAQPi4 and

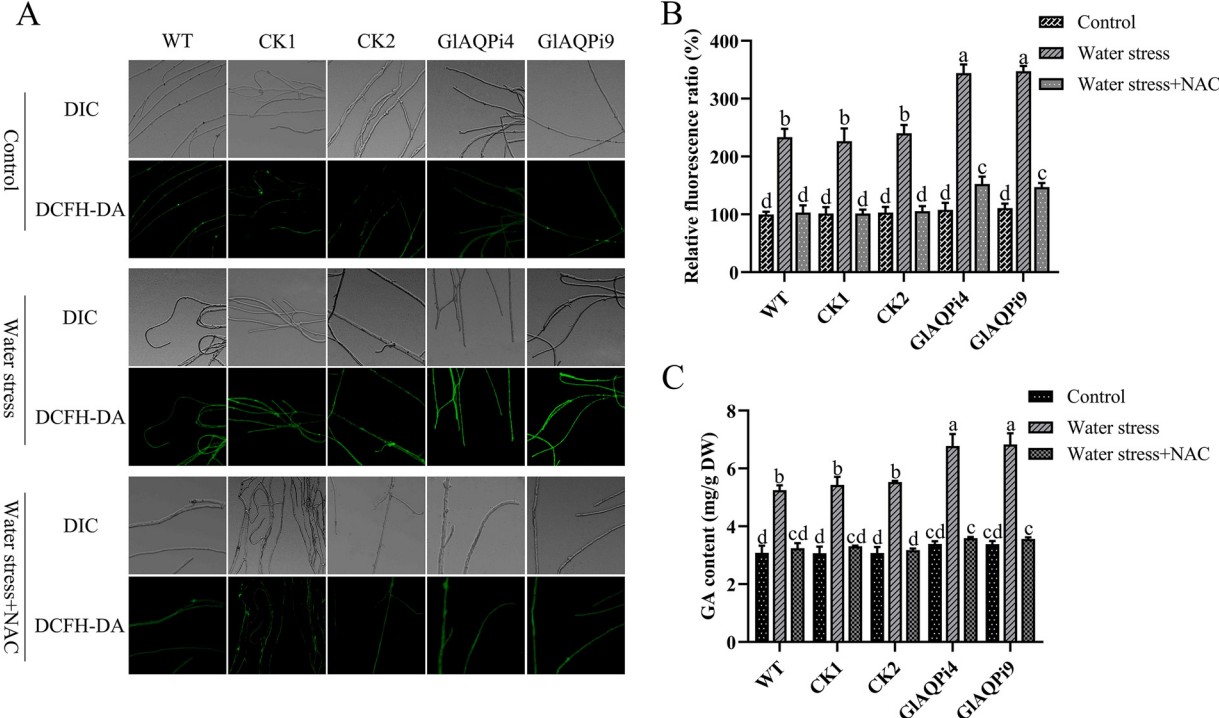

**FIG 4** GlAQP affects GA biosynthesis via ROS under water stress. (A) Change in ROS level detected by DCFH-DA staining in *GlAQP*-silenced strains treated with 0.5 mM of the ROS scavenger NAC under water stress. (B) Change in ROS fluorescence ratio in *GlAQP*-silenced strains treated with 0.5 mM of the ROS scavenger NAC under water stress. (C) The GA content in *GlAQP*-silenced strains treated with 0.5 mM of the ROS scavenger NAC under water stress. The values indicate the mean ± SD of three independent experiments. Different letters indicate significant differences between treatments (Duncan's multiple range test, $P < 0.05$).

GlAQPi9 increased by 53.64% and 51.78%, respectively (Fig. S5A). In addition, after *GlAQP* had been silenced, the activities of the antioxidant enzymes CAT, SOD, and GPx had decreased by 28 to 29%, 26 to 28%, and 29 to 30%, respectively, compared with WT (Fig. S4D to F). These results demonstrate that GlAQP can regulate the intracellular ROS level of *G. lucidum* under water stress.

To further explore the role of GlAQP-regulated ROS in GA biosynthesis under water stress, the ROS scavenger NAC was added and its effect on intracellular ROS level and GA content was assessed. Under water stress, NAC addition decreased the ROS levels of GlAQPi4 and GlAQPi9 by 52.95% and 55.26%, respectively (Fig. 4A and B); the $H_2O_2$ contents by 43.75% and 42.91%, respectively (Fig. S5B); and the GA contents by 46.98% and 47.82%, respectively (Fig. 4C). NAC addition also significantly downregulated the transcription level of key enzyme genes in the GA biosynthetic pathway of *GlAQP*-silenced strains (Fig. S4G to I). These results indicate that GlAQP regulates GA biosynthesis under water stress via ROS.

**The effects of GlAQP on ROS level and GA content differ between early and late stages of water stress.** Because aquaporin can transport $H_2O_2$ (27), the mechanism with which it regulates ROS was investigated. As shown in Fig. S6A to C, the ROS level and $H_2O_2$ content in WT were low and slowly increased during the early stage of water stress. However, during the late stage of water stress, both sharply increased to a high level. Therefore, water stress was divided into an early stage (1.5 days) and a late stage (3 days) to study the effect of GlAQP on ROS levels.

The intracellular ROS levels and $H_2O_2$ contents of *GlAQP*-silenced strains at early and late stages of water stress were assessed. The results show that at the early stage of water stress, the intracellular ROS levels of GlAQPi4 and GlAQPi9 were 16.11% and 17.10% lower than in WT, respectively, and $H_2O_2$ contents were 14.77% and 15.01% lower, respectively. At the late stage of water stress, the intracellular ROS levels of GlAQPi4 and GlAQPi9 were 35.86% and 34.54% higher, and $H_2O_2$ contents were 54.82% and 53.79% higher than WT,

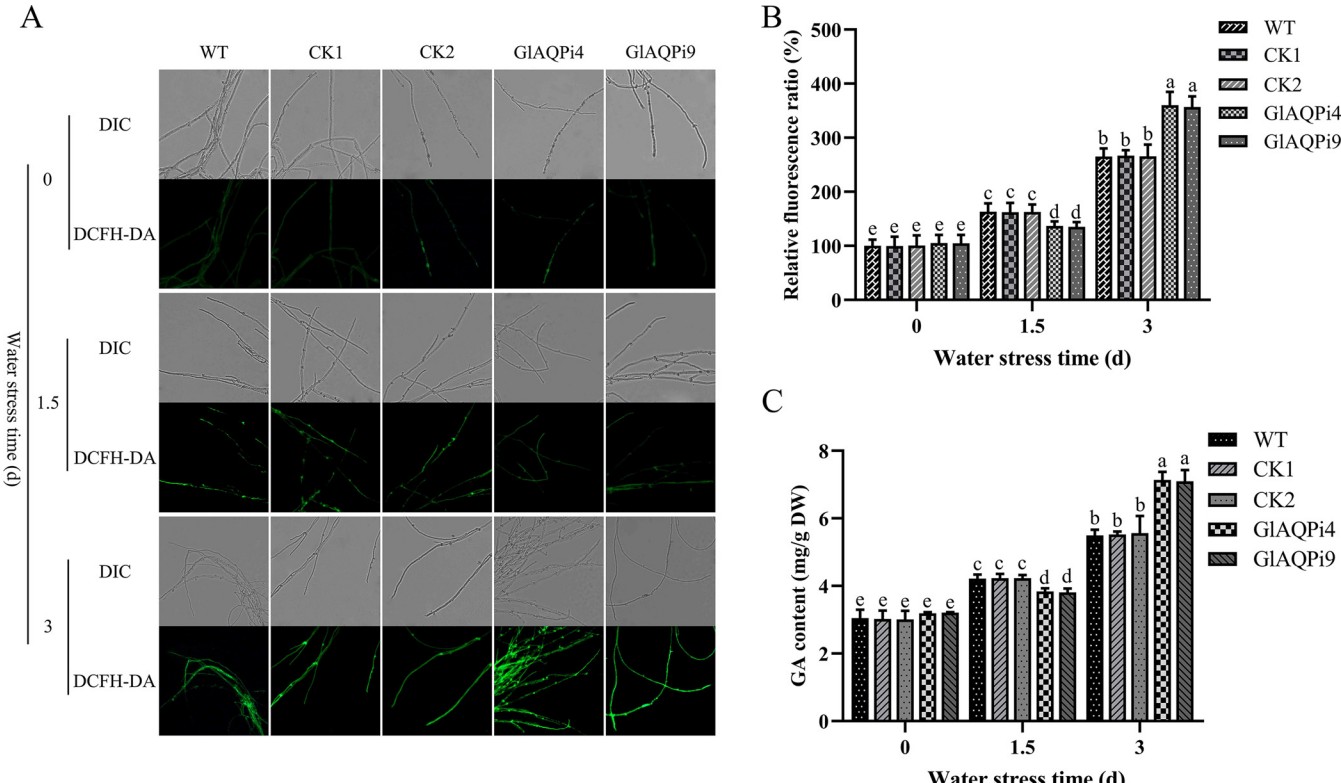

**FIG 5** Effect of silencing *GlAQP* on ROS level and GA content of *G. lucidum* under water stress at early and late stages. (A) Change in ROS level detected by DCFH-DA staining in WT, CK and *GlAQP*-silenced strains of *G. lucidum* under water stress at early and late stages. (B) Change in ROS fluorescence ratio in WT, CK, and *GlAQP*-silenced strains of *G. lucidum* under water stress at early and late stages. (C) The GA content in WT, CK, and *GlAQP*-silenced strains of *G. lucidum* under water stress at early and late stages. The values indicate the mean ± SD of three independent experiments. Different letters indicate significant differences between treatments (Duncan's multiple range test, $P < 0.05$).

respectively (Fig. 5A and B; Fig. S5C). GA contents of GlAQPi4 and GlAQPi9 were 9.02% and 9.65% lower than in WT at the early stage of water stress, while at the late stage GlAQPi4 and GlAQPi9 were 29.83% and 29.14% higher (Fig. 5C). The transcript levels of *hmgr*, *sqs*, and *osc* in GlAQPi4 and GlAQPi9 were all lower than those in WT under early stage water stress but at the late stage all were higher (Fig. S7A to C).

To further study the regulatory role of GlAQP in ROS level and GA biosynthesis under water stress at both early and late stages, the *GlAQP*-overexpressing strains OE::GlAQP-11 and OE::GlAQP-19 were constructed. The transcription levels of *GlAQP* in these two strains were 3.26 and 3.23 times that of WT, respectively (Fig. S8A). The tolerance of *GlAQP*-overexpressing strains to water stress was higher than that of the WT strain. Compared with WT, the growth inhibition rates of OE::GlAQP-11 and OE::GlAQP-19 decreased by 39.90% and 40.91%, respectively (Fig. S8B and S8C). At the early stage of water stress, ROS levels in OE:: GlAQP-11 and OE::GlAQP-19 were 23.86% and 21.58% higher, and $H_2O_2$ contents were 34.24% and 31.80% higher than in WT, respectively. At the late stage of water stress, the ROS levels of these two strains were 15.86% and 16.54% lower, and $H_2O_2$ contents were 13.51% and 15.23% lower than in WT, respectively (Fig. 6A and B; Fig. S5D). In addition, GA contents in both overexpressing strains were 12.50% and 11.97% higher than that in WT under early stage water stress and 9.03% and 8.80% lower under late-stage water stress (Fig. 6C). At the same time, the transcription levels of *hmgr*, *sqs*, and *osc* in OE::GlAQP-11 and OE::GlAQP-19 strains were higher than those of WT under water stress at early stage but lower at the late stage (Fig. S7D to F). These results indicate that the regulatory effects of GlAQP on the GA content via ROS differed between early and late stages of water stress.

**Cross talk between GlAQP and NOX modulates the effects of ROS balance on GA biosynthesis of *G. lucidum* under water stress.** NOX is an important enzyme catalyzing ROS production in *G. lucidum* (21). NOX activity increased significantly under

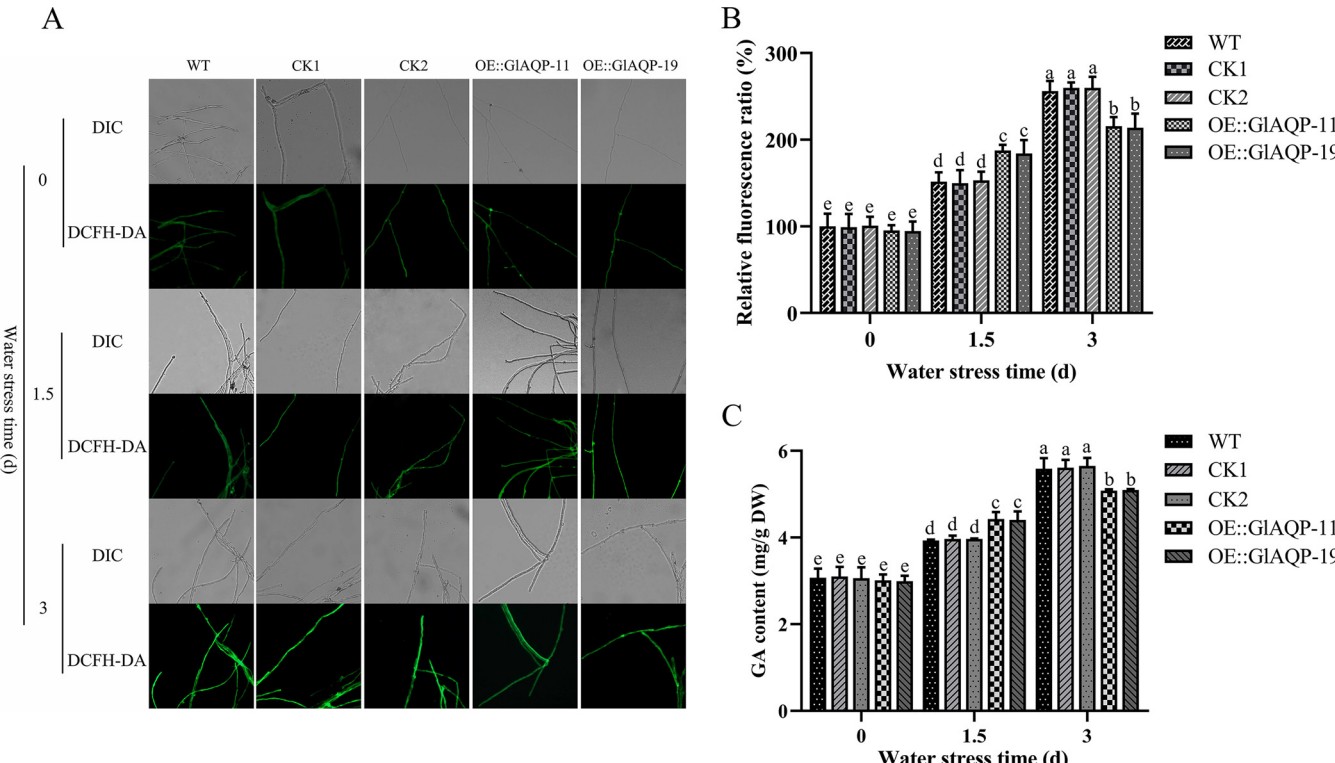

**FIG 6** Effect of overexpressing *GlAQP* on ROS level and GA content of *G. lucidum* under water stress at early and late stages. (A) Change in ROS level detected by DCFH-DA staining in WT, CK, and *GlAQP*-overexpressing strains of *G. lucidum* under water stress at early and late stages. (B) Change in ROS fluorescence ratio in WT, CK, and *GlAQP*-overexpressing strains of *G. lucidum* under water stress at early and late stages. (C) The GA content in WT, CK, and *GlAQP*-overexpressing strains of *G. lucidum* under water stress at early and late stages. The values indicate the mean ± SD of three independent experiments. Different letters indicate significant differences between treatments (Duncan's multiple range test, $P < 0.05$).

water stress, and ROS regulate GA biosynthesis under water stress. Therefore, the NOX inhibitor diphenyleneiodonium chloride (DPI) was added, and previously constructed *NOX*-silenced strains (NOXABi6, NOXABi10, NOXRi4, and NOXRi7) were used to study the role of NOX on ROS level and GA biosynthesis in *G. lucidum* under water stress. Compared with WT, the ROS levels in these four *NOX*-silenced strains decreased by 50.37%, 50.87%, 51.61%, and 51.32%, respectively, and $H_2O_2$ contents decreased by 56.21%, 55.76%, 53.98%, and 52.45%, respectively. In WT, addition of DPI decreased the intracellular ROS level by 48.84% and the $H_2O_2$ content by 51.23% (Fig. 7A and B; Fig. S9A). Compared with WT, the GA content in the four *NOX*-silenced strains decreased by 35.04%, 34.09%, 33.90%, and 33.45%, respectively. The addition of DPI decreased the GA content by 34.88% in WT (Fig. 7C). The transcription level of key enzyme genes of the GA biosynthetic pathway was significantly downregulated after the addition of 10 $\mu$M DPI and in *NOX*-silenced strains to WT (Fig. S9B to D). These results indicate that ROS produced by NOX participate in the biosynthesis of GA under water stress.

It has been reported that AQP can transport $H_2O_2$ produced by NOX into cells to regulate redox signaling (28). The results presented above suggest that GlAQP has different regulatory effects on ROS level depending on whether cells experience early or late stages of water stress. Therefore, the mutual influence of GlAQP and NOX under water stress was studied at both early and late stages. The results showed that NOX activities in GlAQPi4 and GlAQPi9 were 32.69% and 33.03% higher than in WT, respectively. The transcription levels of *NOXA*, *NOXB*, and *NOXR* were all significantly increased at the early stage of water stress in *GlAQP*-silenced strains compared with WT. Under late-stage water stress, the NOX activities in GlAQPi4 and GlAQPi9 were 19.55% and 18.55% lower than in WT, respectively, and the transcription levels of *NOXA*, *NOXB*, and *NOXR* significantly decreased in *GlAQP*-silenced strains (Fig. 8A; Fig. S10). The transcription level of *GlAQP* increased significantly in

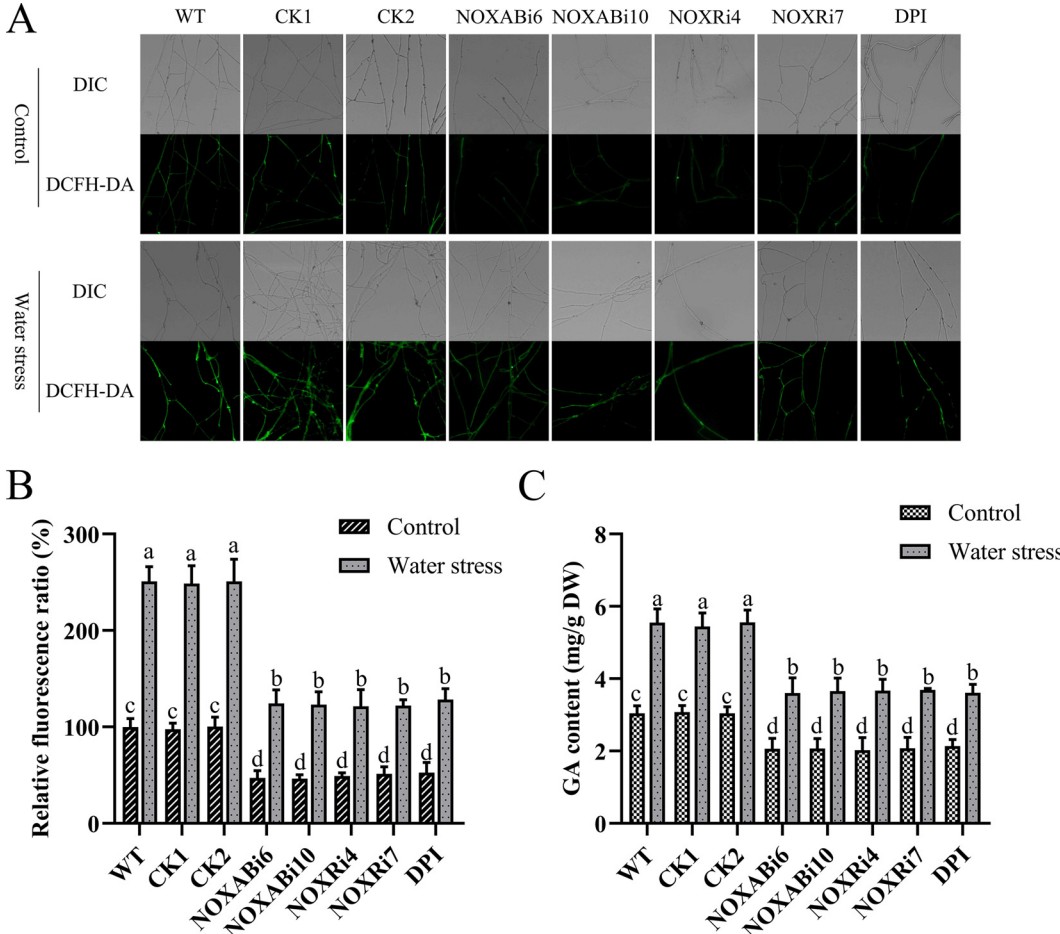

**FIG 7** Effect of silencing *NOX* and adding DPI on ROS level and GA content of *G. lucidum* under water stress. (A) Change in ROS level detected by DCFH-DA staining in *NOX*-silenced strains and 10 μM DPI-treated WT under water stress. (B) Change in ROS fluorescence ratio in *NOX*-silenced strains and 10 μM DPI-treated WT under water stress. (C) The GA content in *NOX*-silenced strains and 10 μM DPI-treated WT under water stress. The values indicate the mean ± SD of three independent experiments. Different letters indicate significant differences between treatments (Duncan's multiple range test, $P < 0.05$).

*NOX*-silenced strains and the DPI-treated strain at the early stage of water stress, while its transcription level decreased significantly at the late stage (Fig. 8B). These results indicate that there was cross talk between GlAQP and NOX at both early and late stages of water stress.

## DISCUSSION

Water stress is one of the main environmental stresses microorganisms may face, which has an important impact on the growth, development, morphology, physiological activities, and metabolism of microorganisms. The results of this study suggest that water stress inhibited the growth of *G. lucidum* while increasing GA biosynthesis. Many signaling molecules play important roles when microorganisms face water stress, among which ROS can mediate rapid systemic signal transduction and activate adaptive pathways. Water stress can lead to a rapid increase of ROS in plants, which in turn causes oxidative stress (29). This increase of ROS caused by stress has been identified as stress signaling (8). Previous studies have demonstrated that heat-stress-induced ROS act as a signal regulating the mycelial branching of *G. lucidum*, the expression of heat shock proteins, and the biosynthesis of the secondary metabolite GA (23). In this study, compared with the control, the intracellular ROS level of *G. lucidum* increased significantly under water stress. Moreover, the activity of antioxidant enzymes, which

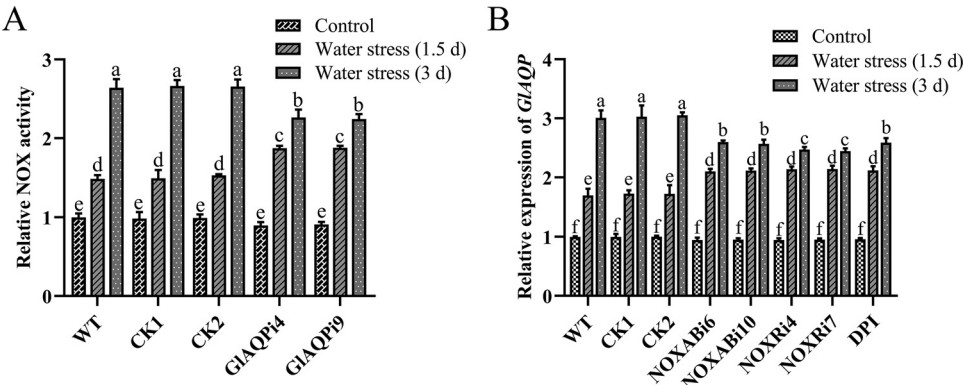

**FIG 8** The mutual influence of GlAQP and NOX under water stress at early and late stages. (A) The activity of NOX in WT, CK, and *GlAQP*-silenced strains of *G. lucidum* under water stress at early and late stages. (B) Transcription level of *GlAQP* in *NOX*-silenced strains and 10 µM DPI-treated WT under water stress at early and late stages. The values indicate the mean ± SD of three independent experiments. Different letters indicate significant differences between treatments (Duncan's multiple range test, $P < 0.05$).

have the function of clearing ROS, also increased significantly under water stress. The increased ROS level under water stress led to an increase in GA biosynthesis. These results indicate that ROS signaling is involved in the GA biosynthesis regulation in *G. lucidum* under water stress. In addition, because water stress can activate ROS in filamentous fungi to participate in signaling transduction, the signaling transduction mechanisms of fungi and plants under water stress are conserved.

AQP can regulate water transport across membranes, which is essential for maintaining water balance and cell vitality. It has been shown that AQP can enhance plant tolerance to water stress (30). This study showed that the AQP gene *GlAQP* of *G. lucidum* was upregulated in response to water stress. After *GlAQP* silencing, the tolerance of *G. lucidum* to water stress and oxidative stress decreased significantly. These results indicate that GlAQP has a positive effect on *G. lucidum* coping with water stress. Moreover, after *GlAQP* silencing, the intracellular ROS level and GA content of *G. lucidum* increased significantly, while the activity of antioxidant enzymes decreased significantly. Similar findings have been reported in plants. Zhou et al. (26) found that the wheat AQP gene *TaAQP7* can improve water retention capacity, reduce ROS accumulation, and enhance the activity of antioxidant enzymes, thereby conferring drought tolerance to genetically modified tobacco. In addition, overexpression of the wheat AQP gene *Td PIP2;1* enhanced the ROS-scavenging ability of wheat by increasing the activity of antioxidant enzymes under water stress, thereby enhancing drought tolerance (31).

The function of AQP has also been studied in other filamentous fungi. AQP8 can regulate ROS production in *Botrytis cinerea*, which in turn affects its infection ability (32). In *F. graminearum*, the absence of AQP1 led to an increase in deoxynivalenol content (33). ROS level and GA content decreased significantly after adding the ROS scavenger NAC. These results indicate that GlAQP may regulate the redox balance under water stress by affecting the activity of antioxidant enzymes, thereby regulating the secondary metabolism of *G. lucidum*. Overall, in filamentous fungi, AQP can regulate both ROS levels and secondary metabolism.

The results show that GlAQP has different effects on ROS levels and GA content at early and late stages of water stress. At the early stage, *GlAQP* silencing decreased both the intracellular ROS level and GA content, while *GlAQP* overexpression increased the ROS level and GA content. However, the phenomenon was the opposite at the late stage of water stress. This may be because GlAQP can promote the bidirectional transmembrane transport of $H_2O_2$ (34). AQP can transport extracellular $H_2O_2$ into cells and mediate signal transduction. Chemokine-dependent T-cell migration is dependent on the uptake of $H_2O_2$ by AQP3, which is indispensable for immune responses (35). In rats,

AQP5 promotes extracellular $H_2O_2$ uptake, improves cell survival, and increases resistance to oxidative stress (36). AQP can also promote the excretion of $H_2O_2$ from cells and reduce oxidative damage. $H_2O_2$ is constantly generated in mitochondria by aerobic metabolism, and mitochondrial Aqp8b-mediated $H_2O_2$ efflux is essential for many vital processes such as detoxification, ATP production, and motility of teleost sperm (37). A recent study on *Streptococcus* found that $H_2O_2$ induced the expression of the AQP gene. AQP can promote the bidirectional transmembrane transport of $H_2O_2$ and detoxify by rapidly exporting $H_2O_2$, thus protecting *Streptococcus* from oxidative stress caused by $H_2O_2$ (38). GlAQP may be able to transport extracellular $H_2O_2$ into the cell to participate in ROS signal transduction and biosynthesis of secondary metabolites under early stage water stress. Then, at the late stage, $H_2O_2$ is expelled to remove excess ROS and detoxify and regulate the biosynthesis of secondary metabolites.

As an important source of ROS localized on the cell membrane, NOX plays an important role in the processes of ROS generation and signal transduction. Research on tomatoes showed that NOX actively regulates disease resistance to *B. cinerea* and tolerance to water stress (39). Another study found that NOX activity increased under water stress and participated in the response of rice to water stress (40). AQP plays an important role in $H_2O_2$ produced by NOX-entering cells. AQP3 mediates the uptake of $H_2O_2$ produced by NOX into mammalian cells to regulate downstream intracellular signal transduction (41). AQP8 transports $H_2O_2$ generated by NOX2 across the plasma membrane to promote signaling in B cells (42). The present study showed that water stress induced NOX activity in *G. lucidum*. In both *NOX*-silenced strains and the DPI-treated strain, ROS level and GA content were significantly decreased in response to water stress. These results indicate that ROS generated by NOX regulated GA biosynthesis under water stress. Moreover, cross talk was found between GlAQP and NOX at both early and late stages of water stress. Compared with WT, the transcription levels of *NOXA*, *NOXB*, and *NOXR,* and the activities of NOX in *GlAQP*-silenced strains increased significantly. The transcription level of *GlAQP* in *NOX*-silenced strains and the DPI-treated strain also increased significantly at the early stage of water stress. However, the transcription level of *NOXA*, *NOXB*, and *NOXR* as well as the activity of NOX in *GlAQP*-silenced strains were significantly decreased. The transcription level of *GlAQP* in *NOX*-silenced strains and the DPI-treated strain had also decreased compared with WT at the late stage of water stress.

At the early stage of water stress, *G. lucidum* needs the production of $H_2O_2$ by NOX as a signal to regulate adaptations to water stress. $H_2O_2$ transport was blocked after *GlAQP* was silenced. The intracellular $H_2O_2$ content was low, resulting in low GA content. The transcription levels of *NOXA*, *NOXB*, and *NOXR* and the activity of NOX increased, which increased the production of $H_2O_2$. After *NOX* was silenced or when DPI was added at the early stage of water stress, the produced $H_2O_2$ content was insufficient. The transcription level of *GlAQP* was enhanced to increase the absorption of $H_2O_2$. Under the late stage of water stress, the intracellular ROS of *G. lucidum* showed an explosive state. GlAQP performed the function of transporting $H_2O_2$ out of the cell to reduce the oxidative damage the cell experiences. After *GlAQP* silencing, intracellular $H_2O_2$ cannot be transported out of the cell, resulting in excessive intracellular ROS level and GA content. Therefore, the transcription levels of *NOXA*, *NOXB*, and *NOXR* and the activity of NOX decreased to reduce the production of ROS. When *NOX* was silenced or DPI was added, the $H_2O_2$ produced by NOX was reduced, which led to a decrease in ROS levels. The transcription level of *GlAQP* was reduced correspondingly to reduce ROS excretion from the cell.

In summary, the results of this study showed that in *G. lucidum* water stress leads to increases in ROS level and GA content. ROS regulates GA biosynthesis under water stress. The expression of the AQP gene *GlAQP* is induced by water stress. GlAQP is related to the water stress tolerance of *G. lucidum*, and under water stress, it influences the biosynthesis of secondary metabolites via ROS. In the early and late stages of water stress, GlAQP has opposite effects on the ROS level and GA biosynthesis. Moreover,

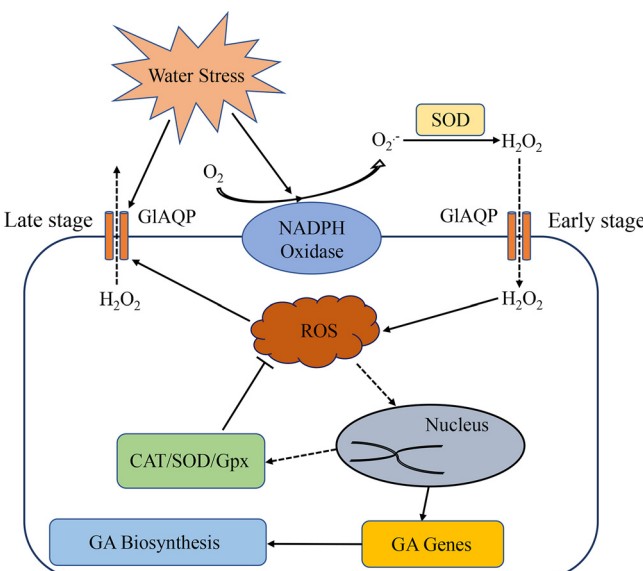

**FIG 9** Proposed schematic representation of cross talk between GlAQP and NOX modulate effects of ROS balance on GA biosynthesis of *G. lucidum* under water stress. The activity of NOX increases under water stress, which in turn produces more $H_2O_2$ as ROS signaling to regulate GA biosynthesis. On the other hand, water stress activates the expression of *GlAQP*, which may have the function of transporting $H_2O_2$. Under water stress at early stage, GlAQP transports the extracellular $H_2O_2$ produced by NOX into the cell in response to water stress and to regulate GA biosynthesis. Under water stress at late stage, GlAQP transports excess intracellular ROS out of the cell, thereby regulating GA biosynthesis.

NOX participates in ROS production and GA biosynthesis. Cross talk between GlAQP and NOX regulates GA biosynthesis via ROS at both early and late stages of water stress (Fig. 9). This research provides information on the biological response mechanism of *G. lucidum* to water stress. New insights on the cell signaling cascade of *G. lucidum* tolerance to water stress are provided that also incorporate the biosynthesis of secondary metabolites involved in NOX and GlAQP.

## MATERIALS AND METHODS

**Strains and culture conditions.** The *G. lucidum* strain ACCC53264 was obtained from the China Agricultural Culture Collection as a parent strain and grown at 28°C in CYM medium (1% maltose, 2% glucose, 0.2% yeast extract, 0.2% tryptone, 0.05% $MgSO_4 \cdot 7H_2O$, and 0.46% $KH_2PO_4$). The *NOX*-silenced strains were constructed as previously (43).

**Water stress and chemical treatment.** The medium containing PEG was prepared according to the method described by Verslues et al. (25). To measure the effect of water stress on the growth of *G. lucidum* mycelium, *G. lucidum* was inoculated on CYM solid medium containing PEG and cultivated for 7 days. For the fermentation experiment in liquid medium containing PEG, *G. lucidum* was first cultivated in CYM liquid medium at 28°C and 150 rpm for 4 days. Then, the mycelia were collected, rinsed with sterile water, transferred to CYM liquid medium containing PEG, and cultured under the same conditions for 3 days. After fermentation, mycelia were collected for RNA extraction, as well as determination of enzyme activity and GA content. For experiments involving NAC, VC, $H_2O_2$, and DPI treatments, 0.5 mM NAC, 1 mM VC, 8 mM $H_2O_2$, and 10 $\mu$M DPI were added to the medium, respectively.

**Cloning and bioinformatics analysis of the *GlAQP* gene.** Because of the similarity between the amino acid sequences of *Saccharomyces cerevisiae* AQP (ADC55558.1) and GlAQP, a BLAST search was performed in the *G. lucidum* genome database (18) to identify the AQP gene in *G. lucidum*. *G. lucidum* cDNA was used as the template, and the entire *GlAQP* sequence of *G. lucidum* was amplified by PCR using the primers listed in Table S1. The fragment was inserted into the pMD19-T vector (TaKaRa, Dalian, China) for sequencing. The positions of open reading frames and exons/introns were identified by comparing genome and cDNA sequences. The conserved domains in the GlAQP protein were identified using the NCBI Conserved Domain Database (http://www.ncbi.nlm.nih.gov/cdd) and the HMMTOP program 2.0 (http://www.enzim.hu/hmmtop/index.php). Subcellular localization prediction was conducted by Euk-mPLoc 2.0 (http://www.csbio.sjtu.edu.cn/bioinf/euk-multi-2). The DNAMAN program was used for multiple sequence alignments, and a phylogenetic tree was generated using MEGA 7.0.

**Construction of RNAi and overexpression of plasmids and strains.** The fungal RNAi-silencing vector pAN7-dual was constructed (20). The dual-promoter-silencing vector driven by the glyceraldehyde-3-phosphate dehydrogenase (gpd) promoter and 35S promoter was used to suppress the expression of

*GlAQP*. The coding region of *GlAQP* was amplified using the primers listed in Table S1. The RNAi-silencing vector pAN7-dual-GlAQPi was transformed into *G. lucidum* according to a previously described method (20).

Overexpressing strains were constructed using the pGL-GPE vector. The *egfp* gene in pGL-GPE was replaced with the *GlAQP* full-length fragment using BamH I and Xba I double restriction enzyme digestion. The new plasmid was designated pGL-*GlAQP*. The transformation of *G. lucidum* with pGL-*GlAQP* was conducted using the *Agrobacterium tumefaciens*-mediated transformation method (19).

After transformation, transformants were selected on CYM solid medium containing 100 $\mu$g/mL hygromycin B. The silencing/overexpressing efficiencies of transformants were analyzed using real-time PCR. Two transformants with high-silencing/overexpressing efficiencies were selected for further research.

**Real-time PCR analysis of gene expression.** Quantitative real-time PCR was used to evaluate the expression levels of gene-specific mRNA. The housekeeping gene 18S rRNA was selected as reference gene to standardize gene expression (43). The relative expression levels of genes were calculated by the $2^{-\Delta\Delta CT}$ method (44). Gene fragments were amplified by real-time PCR using primers listed in Table S1.

**ROS detection method.** ROS levels were detected as previously described (43). Mycelia were stained with DCFH-DA for 20 min and then observed under a Zeiss Axio Imager A1 fluorescence microscope (Zeiss, Wetzlar, Germany). The average fluorescence intensity of DCFH-DA in mycelia was analyzed using ZEN 2012 lite (Zeiss software) (43). A commercial $H_2O_2$ determination kit (Beyotime, Shanghai, China) was used to measure the $H_2O_2$ content of samples.

**Determination of enzyme activity.** CAT activity was determined by measuring the decomposition rate of $H_2O_2$ at 240 nm (45). One unit of SOD activity was defined as the amount of enzyme required to inhibit the photoreduction of nitro blue tetrazolium by 50%, as detected at 560 nm (46). GPx activity (nmol NADPH/min mL) was measured in the supernatant using a cellular glutathione peroxidase assay kit (Beyotime, Shanghai, China) that measures the coupled oxidation of NADPH during glutathione reductase recycling of oxidized glutathione from GPx-mediated reduction of t-butyl peroxide. NOX activity was detected using a NOX activity kit (Solarbio, China). NOX oxidizes NADH to NAD, while the blue 2,6-dichlorophenol indigo (DCPIP) is reduced to colorless DCPIP. The NADH oxidase activity was obtained by measuring the reduction rate of blue DCPIP at 600 nm.

**Detection of GA content.** GA was extracted from *G. lucidum* mycelia and measured according to the previously described method (21).

**Growth susceptibility assay.** To evaluate the sensitivity of WT, CK (silenced empty vector control strain), and *GlAQP*-silenced strains to water stress or oxidative stress, a *G. lucidum* mycelial tip plug with a diameter of 6 mm was inoculated on CYM solid medium containing PEG or $H_2O_2$. This plug was cultured at 28°C for 7 days, after which the diameter of each colony was measured. The growth inhibition rate of WT was arbitrarily set to 1, to which the growth inhibition rates of other strains were compared.

**Statistical analysis.** All experiments were repeated at least three times, and the values represent the mean $\pm$ SD of three independent experiments. The data were analyzed using Student's *t* test or Duncan's multiple range test. $P < 0.05$ was considered statistically significant.

## SUPPLEMENTAL MATERIAL

Supplemental material is available online only.

**SUPPLEMENTAL FILE 1**, PDF file, 2.3 MB.

## ACKNOWLEDGMENTS

This work was supported by the National Natural Science Foundation of China (31871782), the Project of Sanya Yazhou Bay Science and Technology City (grant number SCKJ-JYRC-2022-68), the Guidance Foundation, the Sanya Institute of Nanjing Agricultural University (grant number NAUSY-MS17), and the earmarked fund for China Agriculture Research System (project number CARS20).

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
