## [Reviewer comments · Microbiology Spectrum]

Microbiology Spectrum

Cross Talk between GIAQP and NOX Modulates the Effects of ROS Balance on Ganoderic Acid Biosynthesis of *Ganoderma lucidum* under Water Stress

Quanyu Zhu, Ang Ren, Juan Ding, Jian He, Mingwen Zhao, Ai-Liang Jiang, Xiaolin Zhou, Jieying Wang, and Qin He

Corresponding Author(s): Qin He, Nanjing Agricultural University

Review Timeline:

Submission Date:	April 7, 2022
Editorial Decision:	June 20, 2022
Revision Received:	August 17, 2022
Editorial Decision:	September 2, 2022
Revision Received:	September 25, 2022
Accepted:	October 2, 2022

Editor: Matthew Anderson

Reviewer(s): Disclosure of reviewer identity is with reference to reviewer comments included in decision letter(s). The following individuals involved in review of your submission have agreed to reveal their identity: Yejun Han (Reviewer #2); Xuan-Wei Zhou (Reviewer #4)

Transaction Report:

DOI: <https://doi.org/10.1128/spectrum.01297-22>

June 20, 2022

Dr. Qin He
Nanjing Agricultural University
College of Life Sciences
Nanjing, Jiangsu
China

Re: Spectrum01297-22 (Cross Talk between GIAQP and NOX Regulates the Ganoderic Acid Biosynthesis of *Ganoderma lucidum* via ROS under Water Stress)

Dear Dr. Qin He:

After having been reviewed, the manuscript has some important scientific points to be made but needs some additional work to be more informative for readers. Please address comments by Reviewers 2 and 4. Reviewer 3's comments can be largely ignored.

Link Not Available

Sincerely,

Matthew Anderson

Journals Department
Reviewer comments:

Reviewer #2 (Comments for the Author):

The manuscript authored by et al. studied ganoderic acid synthesis in *Ganoderma lucidum* under water stress condition. The production of ganoderic acid was first studied in *Ganoderma lucidum* cultured under different water stress condition. The NADPH oxidase, reactive oxygen species, transcription of aquaporin gene GIAQP, and gene GIAQP content in *Ganoderma lucidum* were further analyzed. The work is interesting and enriches the understanding of ganoderic acid production in *Ganoderma lucidum*. The following are comments for consideration.

- 1: The title is a bit vague and I would recommend reconsideration.
- 2: Abstract, line 19-29, it's better to re-organize the section to make it clearer.
- 3: The Importance section is not properly written, don't repeat the results as that of abstract, please check the instruction and reorganize.
- 4: Line 117-119, "The results showed that with the increase of PEG concentration, the GA content showed a trend of first increasing and then decreasing.", please explain why "first increasing and then decreasing"? In addition, the difference in GA content can be statistically analyzed.
- 5: Line 269, Cross talk between GIAQP and NOX under water stress at early and late stages, I would recommend to rename the section and express the content or highlight directly.
- 6: Figure 3-8, the contents of the figures can be simplified, and switch unneeded parts to supplementary file.
- 7: Figure 9, since the regulatory process of GA synthesis has not been fully studied, it might be titled "Proposed schematic representation of cross talk between GIAQP and NOX regulates the GA biosynthesis of *G. lucidum* via ROS under water stress".

Reviewer #3 (Comments for the Author):

More statistical analysis are needed.
Gen expression is low.

Reviewer #4 (Comments for the Author):

In this manuscript, the authors showed that water stress led to an increase in ROS level and GA content in *G. lucidum*. Then, they identified a AQP gene GIAQP, and discussed its roles in the process of *G. lucidum* coping with water stress and GA biosynthesis. At last, cross talk between GIAQP and NOX under water stress was discussed. Still, there are several issues that need to be improved before publication.

1. The manuscript indicated that GIAQP has a positive effect on *G. lucidum* coping with water stress using GIAQP-silencing strain. Whether GIAQP-overexpressing strain is needed to display the corresponding results.
2. The representation of significance of the difference should be in a unified form in figures.
3. When the abbreviation appears for the first time in the text, its full name should be noted.
4. In Fig. S5A, means of the numbers should be indicated.
5. In Fig. 9, "Nox Complex" is unclear.
6. The contents of chemicals such as H₂O₂ should be added in figure legends.

Staff Comments:

Preparing Revision Guidelines

Please return the manuscript within 60 days; if you cannot complete the modification within this time period, please contact me. If you do not wish to modify the manuscript and prefer to submit it to another journal, please notify me of your decision immediately so that the manuscript may be formally withdrawn from consideration by Microbiology Spectrum.

In this manuscript, the authors showed that water stress led to an increase in ROS level and GA content in *G. lucidum*. Then, they identified a AQP gene GIAQP, and discussed its roles in the process of *G. lucidum* coping with water stress and GA biosynthesis. At last, cross talk between GIAQP and NOX under water stress was discussed. Still, there are several issues that need to be improved before publication.

1. The manuscript indicated that GIAQP has a positive effect on *G. lucidum* coping with water stress using GIAQP-silencing strain. Whether GIAQP-overexpressing strain is needed to display the corresponding results.

2. The representation of significance of the difference should be in a unified form in figures.

3. When the abbreviation appears for the first time in the text, its full name should be noted.

4. In Fig. S5A, means of the numbers should be indicated.

5. In Fig. 9, "Nox Complex" is unclear.

6. The contents of chemicals such as H₂O₂ should be added in figure legends.

Authors demonstrated that the regulation of GIAQP and NOX in GA biosynthesis via ROS under water stress. This is an interesting and solid work unveiling mechanisms of GA biosynthesis through Cross Talk between AQP and NOX under water stress. The language in the manuscript needs polishing. I feel that this article is suitable to be published in the journal after minor revision.

Responses to the reviewers' concerns

August 17, 2022

Dear editor Matthew Anderson,

Thanks for giving us a chance to revise the paper, we are very grateful to you and the reviewers for the comments on our manuscript entitled “**Cross Talk between GIAQP and NOX Regulates the Ganoderic Acid Biosynthesis of *Ganoderma lucidum* via ROS under Water Stress**” (ID: Spectrum01297-22). These comments are very valuable and helpful for revising and improving our paper. We have studied these comments carefully. Point-by-point responses to the comments and specific issues are shown below in square brackets.

Thanks for all the help.

Best wishes,

Sincerely,

Qin He

Responses to reviewer #2's comments

Reviewer #2: The manuscript authored by et al. studied ganoderic acid synthesis in *Ganoderma lucidum* under water stress condition. The production of ganoderic acid was first studied in *Ganoderma lucidum* cultured under different water stress condition. The NADPH oxidase, reactive oxygen species, transcription of aquaporin gene *GIAQP*, and gene *GIAQP* content in *Ganoderma lucidum* were further analyzed. The work is interesting and enriches the understanding of ganoderic acid production in *Ganoderma lucidum*. The following are comments for consideration.

1: The title is a bit vague and I would recommend reconsideration.

[Answer: Thanks for the reviewer's suggestion.

We have reconsidered the title and revised it as “Cross Talk between GIAQP and NOX Modulate Effect of ROS Balance on Ganoderic Acid Biosynthesis of *Ganoderma lucidum* under Water Stress”. Because cross talk between GIAQP and

NOX positively regulates the GA biosynthesis of *G. lucidum* via ROS under water stress at early stage but negatively regulates the GA biosynthesis via ROS at late stage.]

2: Abstract, line 19-29, it's better to re-organize the section to make it clearer.

[Answer: Thanks for the reviewer's kind remind.

We have made changes in this section. The sentences have been re-organized and made clear.

Changes are highlighted in red in the manuscript, line 17-31.]

3: The Importance section is not properly written, don't repeat the results as that of abstract, please check the instruction and reorganize.

[Answer: Sorry, we are sorry for repeating the results as that of abstract in the importance section.

We have reorganized it and changes are highlighted in red in the manuscript, please see it in line 37-44.]

4: Line 117-119, "The results showed that with the increase of PEG concentration, the GA content showed a trend of first increasing and then decreasing.", please explain why "first increasing and then decreasing"? In addition, the difference in GA content can be statistically analyzed.

[Answer: Thanks for the reviewer's suggestion.

This may be due to the reduction of water potential in the medium after the addition of PEG, and *Ganoderma lucidum* suffers from water stress. Since GA has the effect of helping *G. lucidum* to resist stress, the content of GA increased under water stress compared with the control. With the increase of PEG concentration, the degree of water stress increased, the biosynthesis of GA also increased. However, when the PEG concentration exceeded 20%, the degree of water stress was too severe, which seriously damaged the growth and metabolism of *G. lucidum*, and the biosynthesis of GA was also seriously damaged, thus, the content of GA began to decrease.

The difference in GA content has been statistically analyzed.

Changes are highlighted in red in the manuscript, line 120-125.]

5: Line 269, Cross talk between GIAQP and NOX under water stress at early and late stages, I would recommend to rename the section and express the content or highlight directly.

[Answer: Thanks for the reviewer's comments, we have renamed the section and express the content.

Changes are highlighted in red in the manuscript, line 278-279.]

6: Figure 3-8, the contents of the figures can be simplified, and switch unneeded parts to supplementary file.

[Answer: Thanks for the reviewer's kind remind.

We have simplified the contents of the figures, and switched unneeded parts to supplementary file. The figures of H₂O₂ content in Fig. 3-6 are combined into Fig. S9 and the figure of H₂O₂ content in Fig. 7 and the figure of the transcription levels of *NOXA*, *NOXB* and *NOXR* in Fig. 8 are combined into Fig. S9]

7: Figure 9, since the regulatory process of GA synthesis has not been fully studied, it might be titled "Proposed schematic representation of cross talk between GIAQP and NOX regulates the GA biosynthesis of *G. lucidum* via ROS under water stress".

[Answer: We are very grateful for the reviewer's suggestion.

We have renamed Figure 9 to "Proposed schematic representation of cross talk between GIAQP and NOX regulates the GA biosynthesis of *G. lucidum* via ROS under water stress".]

Responses to reviewer #4's comments

Reviewer #4: In this manuscript, the authors showed that water stress led to an increase in ROS level and GA content in *G. lucidum*. Then, they identified a *AQP* gene *GIAQP*, and discussed its roles in the process of *G. lucidum* coping with water

stress and GA biosynthesis. At last, cross talk between GIAQP and NOX under water stress was discussed. Still, there are several issues that need to be improved before publication.

1. The manuscript indicated that GIAQP has a positive effect on *G. lucidum* coping with water stress using GIAQP-silencing strain. Whether GIAQP-overexpressing strain is needed to display the corresponding results.

[Answer: Thanks for the reviewer's suggestion.

We are sorry for omitting this important information. We have tested the tolerance of *GIAQP* overexpressed strain to water stress, and the results showed that the tolerance of *GIAQP* overexpressed strains to water stress were higher than that of WT strain. Compared with the WT, the growth inhibition rates of OE::GIAQP-11 and OE::GIAQP-19 were reduced by 39.90% and 40.91% respectively, under water stress, which corresponding to the results of *GIAQP*-silenced strains. The result is shown in supplementary figure S6.

Changes are highlighted in red in the manuscript, line 263-266 and supplemental material.]

2. The representation of significance of the difference should be in a unified form in figures.

[Answer: Thanks for the reviewer's kind remind.

We have unified the representation of significance of the difference in figures.]

3. When the abbreviation appears for the first time in the text, its full name should be noted.

[Answer: Thanks for the reviewer's comments.

We have checked the whole manuscript and revised it.

Changes are highlighted in red in the manuscript.]

4. In Fig. S5A, means of the numbers should be indicated.

[Answer: Thanks for the reviewer's suggestion.

We have indicated the means of the numbers in Fig. S6A.]

5. In Fig. 9, "Nox Complex" is unclear.

[Answer: Sorry, we are sorry for the unclear description.

We have changed "Nox Complex" to "NADPH Oxidase" in Fig. 9.]

6. The contents of chemicals such as H₂O₂ should be added in figure legends.

[Answer: Thanks for the reviewer's suggestion.

We have added the contents of chemicals in figure legends.]

Authors demonstrated that the regulation of GlAQP and NOX in GA biosynthesis via ROS under water stress. This is an interesting and solid work unveiling mechanisms of GA biosynthesis through Cross Talk between AQP and NOX under water stress. The language in the manuscript needs polishing. I feel that this article is suitable to be published in the journal after minor revision.

[Answer: Thanks for the reviewer's kind comment.

We asked an expert in this field to polish the language, also we tried our best to improve the language and revised all the issues you proposed in the revised paper.

We hope the revision is suitable, and the article can be accepted and published in the journal.]

September 2, 2022

Dr. Qin He
Nanjing Agricultural University
College of Life Sciences
Nanjing, Jiangsu
China

Re: Spectrum01297-22R1 (Cross Talk between GIAQP and NOX Modulate Effects of ROS Balance on Ganoderic Acid Biosynthesis of *Ganoderma lucidum* under Water Stress)

Dear Dr. Qin He:

Thank you for taking care to address the reviewer comments provided to you from review. There are two points that we would like to have addressed:

1. Grammar in the manuscript needs to be addressed. There are many instances of grammatical flaws in the manuscript in its current form.
2. The point on line 311-313 of different modes of crosstalk between GIAQP and NOX needs some additional explanation. Both systems respond identically in early and late water stress. IT is therefore unclear what the "different modes of crosstalk" would be since they function in sync throughout the experiment.

As you will see your paper is very close to acceptance. Please modify the manuscript along the lines I have recommended. As these revisions are quite minor, I expect that you should be able to turn in the revised paper in less than 30 days, if not sooner. If your manuscript was reviewed, you will find the reviewers' comments below.

When submitting the revised version of your paper, please provide (1) point-by-point responses to the issues raised by the reviewers as file type "Response to Reviewers," not in your cover letter, and (2) a PDF file that indicates the changes from the original submission (by highlighting or underlining the changes) as file type "Marked Up Manuscript - For Review Only". Please use this link to submit your revised manuscript. Detailed instructions on submitting your revised paper are below.

Link Not Available

Sincerely,

Matthew Anderson

Reviewer comments:

Preparing Revision Guidelines

- Point-by-point responses to the issues raised by the reviewers in a file named "Response to Reviewers," NOT IN YOUR COVER LETTER.

- Upload a compare copy of the manuscript (without figures) as a "Marked-Up Manuscript" file.
- Each figure must be uploaded as a separate file, and any multipanel figures must be assembled into one file.
- Manuscript: A .DOC version of the revised manuscript
- Figures: Editable, high-resolution, individual figure files are required at revision, TIFF or EPS files are preferred

Please return the manuscript within 60 days; if you cannot complete the modification within this time period, please contact me. If you do not wish to modify the manuscript and prefer to submit it to another journal, please notify me of your decision immediately so that the manuscript may be formally withdrawn from consideration by Microbiology Spectrum.

Responses to the editor's concerns

September 25, 2022

Dear editor Matthew Anderson,

Thanks for giving us a chance to revise the paper, we are very grateful to you for the comments on our manuscript entitled “**Cross Talk between GIAQP and NOX Modulate Effects of ROS Balance on Ganoderic Acid Biosynthesis of *Ganoderma lucidum* under Water Stress**” (ID: Spectrum01297-22R1). These comments are very valuable and helpful for revising and improving our paper. We have studied these comments carefully. Point-by-point responses to the comments and specific issues are shown below in square brackets.

Thanks for all the help.

Best wishes,

Sincerely,

Qin He

Responses to the editor's comments

1. Grammar in the manuscript needs to be addressed. There are many instances of grammatical flaws in the manuscript in its current form.

[Answer: Thanks for the editor's suggestion.

We are very sorry for the many grammatical flaws in the manuscript. We have asked a professional language edit service to proofread the manuscript, the grammatical errors in the manuscript have been corrected and attached is the certification of language modification.

In addition, Ailiang Jiang, Xiaolin Zhou and Jieying Wang made great contributions to the content and language modification of the manuscript in these two rounds.

Changes are highlighted in red in the manuscript.]

2. The point on line 311-313 of different modes of crosstalk between GIAQP and NOX needs some additional explanation. Both systems respond identically in early

and late water stress. IT is therefore unclear what the "different modes of crosstalk" would be since they function in sync throughout the experiment.

[Answer: Thanks for the reviewer's kind remind.

We fully agree with the editor's opinion that the modes of cross talk between GIAQP and NOX are consistent at early and late stages of water stress. We have changed it to "These results indicate that there was cross talk between GIAQP and NOX at both early and late stages of water stress".

Changes are highlighted in red in the manuscript, line 313-314.]

October 2, 2022

Dr. Qin He
Nanjing Agricultural University
College of Life Sciences
Nanjing, Jiangsu
China

Re: Spectrum01297-22R2 (Cross Talk between GIAQP and NOX Modulates the Effects of ROS Balance on Ganoderic Acid Biosynthesis of *Ganoderma lucidum* under Water Stress)

Dear Dr. Qin He:

Your manuscript has been accepted, and I am forwarding it to the ASM Journals Department for publication. You will be notified when your proofs are ready to be viewed.

Sincerely,

Matthew Anderson
Editor, Microbiology Spectrum
